# Enhanced Phase Transition Properties of VO_2_ Thin Films on 6H-SiC (0001) Substrate Prepared by Pulsed Laser Deposition

**DOI:** 10.3390/nano9081061

**Published:** 2019-07-24

**Authors:** Xiankun Cheng, Qiang Gao, Kaifeng Li, Zhongliang Liu, Qinzhuang Liu, Qiangchun Liu, Yongxing Zhang, Bing Li

**Affiliations:** School of Physics and Electronic Information, Huaibei Normal University, Huaibei 235000, China

**Keywords:** VO_2_, 6H-SiC, pulsed laser deposition (PLD)

## Abstract

For growing high quality epitaxial VO_2_ thin films, the substrate with suitable lattice parameters is very important if considering the lattice matching. In addition, the thermal conductivity between the substrate and epitaxial film should be also considered. Interestingly, the c-plane of hexagonal 6H-SiC with high thermal conductivity has a similar lattice structure to the VO_2_ (010), which enables epitaxial growth of high quality VO_2_ films on 6H-SiC substrates. In the current study, we deposited VO_2_ thin films directly on 6H-SiC (0001) single-crystal substrates by pulsed laser deposition (PLD) and systematically investigated the crystal structures and surface morphologies of the films as the function of growth temperature and film thickness. With optimized conditions, the obtained epitaxial VO_2_ film showed pure monoclinic phase structure and excellent phase transition properties. Across the phase transition from monoclinic structure (M1) to tetragonal rutile structure (R), the VO_2_/6H-SiC (0001) film demonstrated a sharp resistance change up to five orders of magnitude and a narrow hysteresis width of only 3.3 °C.

## 1. Introduction

Among various vanadium oxides, vanadium dioxide (VO_2_), a narrow bandgap semiconductor with 0.6 eV, has been extensively studied because of its conspicuous metal-insulator transition (MIT) characteristics at the critical phase transition temperature (*T_C_*) of about 68 °C [1]. When the temperature reaches *T_C_*, VO_2_ will change from the monoclinic structure (M1, *P*21/*c*) to tetragonal rutile structure (R, *P*42/*mnm*), accompanied by significant changes in optical and electrical properties [2]. For example, the resistance of VO_2_ will be greatly decreased during the MIT process with the change up to 3–5 orders of magnitude. Due to the peculiar phase change characteristics of VO_2_, which can be triggered by temperatures or some other routes such as electric excitation [3]. VO_2_ shows many promising applications in the field of optoelectronics, such as steep-slope transistors [4], antennas [5] and photoelectronic devices [6].

The performance of VO_2_ based optoelectronic devices is closely associated with the quality of VO_2_ films. As a result, the high quality VO_2_ film preparation is still a focusing point and essential issue. Many previous experiments have been reported about the VO_2_ epitaxial film growth on various substrates such as c-plane sapphire [7,8,9], TiO_2_ [10,11], GaN [12] and ordinary quartz glass [13], directly or with a buffer layer [14,15], For instance, Wang et al. deposited VO_2_ film on p-GaN/Al_2_O_3_ (0001) substrate and explored the heterojunction devices. The growth of VO_2_ films on the ZnO/glass substrate provided a new research direction for the application of electronically controlled smart windows [16]. However, compared with substrates of Si, Al_2_O_3_, GaN, TiO_2_ and glass, which are commonly used to epitaxially grown VO_2_ thin films, SiC has outstanding performance in terms of thermal conductivity [17], electron mobility, thermal stability, and chemical stability. These excellent properties expand the application environment of VO_2_ thin films, such as strong acid and alkali, high temperature and other extreme conditions, while, until now, there have been few reports of epitaxial growth of VO_2_ films on SiC single-crystal substrates.

In this work, VO_2_ thin films were directly grown on 6H-SiC (0001) single-crystal substrates by PLD, the microstructures and electrical properties were also systematically investigated upon the growth temperature and film thickness. Ultra-high quality VO_2_ film with pure monoclinic phase structure and excellent phase transition properties were obtained, which demonstrated a sharp resistance change up to five orders of magnitude and a narrow hysteresis width during the MIT process. The success combination of high quality VO_2_ epitaxial film with the third-generation semiconductor SiC crystal will have an important impact on future devices based on excellent phase transition characteristics**.**

## 2. Materials and Methods

### 2.1. Preparation of VO_2_/6H-SiC Films

The 6H-SiC (0001) single-crystal substrate (double-side polished slice with the resistance of 0.02–0.1 ohm-cm and the size of 5 mm × 5 mm × 0.5 mm, Hefei Yuanjing Technology Materials Co., Ltd.) was firstly ultra-sonic cleaned by acetone solution and deionized water, and then blown dry by pure Nitrogen gas before put into the vacuum growth chamber. A VO_2_ ceramic target (99.99% purity, 25 mm diameter, 2 mm thickness, Zhongnuo New Material (Beijing) Technology Co., Ltd. Haidian, Beijing, China) was fixed at the position perpendicular to the substrate with the distance of 55 mm. VO_2_ films were deposited on the 6H-SiC (0001) substrate by PLD. The pulsed laser (Kr: F laser, λ = 248 nm, 10 Hz) energy density was about 1.6 J/cm^2^. During the deposition and post-annealing process, the oxygen pressure was kept at 4.0 Pa with the flux flow rate of 20 sccm.

In order to explore the optimum growth conditions of VO_2_, the experiment was divided into two groups of S_1_ and S_2_. The samples of S_1_ group were named S_1-1_, S_1-2_, S_1-3_ and S_1-4_, respectively, with corresponding substrate temperatures (deposition time) of 450 °C (30 min), 500 °C (30 min), 550 °C (30 min) and 650 °C (30 min); the samples of S_2_ group were fixed at 500 °C to improve the continuity of the VO_2_ film surfaces by changing the deposition time of 10 min, 30 min, 50 min and 70 min (named S_2-1_, S_2-2_, S_2-3_ and S_2-4_, in turn). All samples were annealed for 20 min after deposition and then naturally cooled to room temperature.

### 2.2. Materials Characterization

The crystal structure of all samples was characterized by X-ray diffractometer (XRD; PANalytical, Empyrean). Field Emission Scanning Electron Microscopy (FE-SEM; HiTACHI Regulus 8220) was used to observe the surface morphologies. X-ray photoelectron spectroscopy (XPS; Thermo, escalab 250XI) was also performed to analyze the chemical states of the obtained samples. All the binding energies were corrected by calibrating the C 1s peak at 284.6 eV. Raman spectra of the VO_2_/6H-SiC films were acquired using the XploRA™ Raman spectrometer (HORIBA Scientific, Ltd. Hefei, Anhui, China), and a 532 nm laser with power of 0.25 mW was used as the excitation source. The resistance versus temperature curve of all samples was measured using a four-probe measurement system with variable temperature ranging from 30 °C to 95 °C.

## 3. Results and Discussion

### 3.1. Influence of Substrate Temperatures and Deposition Time on the Properties of VO_2_/6H-SiC Films

Figure 1 shows the XRD pattern in typical θ–2θ scanning mode for the VO_2_/6H-SiC films. The well-resolved sharp reflection at 35.64°, corresponds to a classic reflections from (0006)-oriented 6H-SiC (PDF#74-1302) substrate [18]. A unique M-phase (020) diffraction peak attributed to VO_2_ at 39.93° was observed, consistent with the report by Liao et al. [19]. No other VO_2_ diffraction peaks were observed in the range of angles tested. According to Figure 1a, the crystallinity of sample S_1-2_ was the best among the four samples in group S_1_. Therefore, it can be inferred that the substrate temperature corresponding to the growth of high quality VO_2_ films on 6H-SiC (0001) was 500 °C. On the basis of the previous experiments, the continuity of the surface of the VO_2_ films was improved by setting different deposition times (10 min, 30 min, 50 min, 70 min). Figure 1b shows that the relative intensity of the VO_2_ (020) diffraction peak at 39.93° increases with increasing deposition time. This indicates that as the deposition time increases, the surface of the VO_2_ film becomes more complete and continuous, which leads to a gradual increase in the quality of the crystal.

As a non-destructive detection method, Raman spectroscopy can quickly analyze and identify the phase of matter. We performed Raman spectroscopy on all samples including the substrate at room temperature and the results are shown in Figure 2. In order to distinguish the characteristic peaks of VO_2_ and 6H-SiC more clearly, we added the Raman spectrum of 6H-SiC in Figure 2a. For the 6H-SiC substrates, three first-order peaks of Si-C vibration were detected at 765, 787 and 965 cm^-1^ corresponding to E2 (TO), E2 (TO) and A1 (LO) mode respectively [20]. The weaker intensity peaks in 6H-SiC are an E2 planar acoustic mode at 147 cm^−1^ [21]. The data in Figure 2a,b show the room temperature M1 phase Raman spectrum of VO_2_/6H-SiC films at 143 (Bg), 191 (Ag), 222 (Ag), 260 (Bg), 307 (Bg), 335 (Ag), 385 (Ag) , 439 (Bg), 497 (Ag) and 612 cm^−1^ (Ag) [22]. Meanwhile, no other peaks were observed, which proved that high quality VO_2_ films can be grown on 6H-SiC substrate. The result is consistent with that of XRD pattern of VO_2_/6H-SiC films.

FE-SEM characterization of all samples was performed to more intuitively investigate the effect of substrate temperature and deposition time on the formation of VO_2_ films. The FE-SEM images of the surface of all samples were presented in Figure 3, and Figure 3a–d FE-SEM images of four different substrate temperature correspond to samples in the S_1_ group. It can be seen that the VO_2_ crystal nucleus on the surface of the 6H-SiC substrate gradually grows with increasing temperature. However, the continuity of the VO_2_ films gradually deteriorates after the substrate temperature exceeds 500 °C. In particular, VO_2_ deposited on the surface of the 6H-SiC substrate formed only a discontinuous island-like particle after the substrate temperature reached 650 °C. This is due to the reflection phenomenon in the process of depositing VO_2_ particles onto the 6H-SiC (0001) substrate, and then becoming more disconnected with an increase in temperature.

Figure 3e–h are FE-SEM images of S_2-1_, S_2-2_, S_2-3_ and S_2-4_ samples in the S_2_ group, respectively. There is no doubt that the number of holes on the surface of the VO_2_/6H-SiC film will gradually decrease as the deposition time increases, which leads to a tighter connection between the VO_2_ grains. This is consistent with the change in the relative intensity of the VO_2_ (020) characteristic peak exhibited by the S_2_ group samples in Figure 1b. In addition, it can be seen from Figure 2b that the Raman characteristic peak of the 6H-SiC substrate weakens or even disappears as the VO_2_ deposition time increases.

It is well known that among the many vanadium oxides (such as VO_2_, VO, V_2_O_3_ and V_2_O_5_), only VO_2_ (M) has the MIT characteristic that appears after heating the sample to the phase transition temperature. Therefore, it is very important to determine the valence state of the V atom in the VO_2_/6H-SiC films for determining the composition of the sample. We studied the precise valence state of vanadium (V) in the sample by XPS. The XPS spectrum of the VO_2_/6H-SiC films is shown in Figure 4a.

In order to more clearly show the approximate composition of the oxide film, we used a high-resolution XPS spectrum centered on O 1_S_ and V 2_P3/2_, as shown in Figure 4b. All elements in the composition can be well identified in the XPS measurement spectrum compared to previously reported literature [23]. The O 1s peak can be curled into two peaks at 530.0 eV and 532.1 eV. And the former van be considered to be related to vanadium oxide and the latter being mainly from hydroxide or carbonate contamination. It is worth noting that the combined energy span of 516.1 eV to 530.0 eV was calculated to be 13.90 eV, which is consistent with previous reports [24]. Meanwhile, the V^4+^ 2_P3/2_ and V^4+^ 2_P1/2_ were located at the binding energies of 516.1 eV and 523.46 eV, [25] respectively. It was confirmed that the valence state of V of the VO_2_/6H-SiC films was mainly composed of V^4+^. Furthermore, the peak at 517.5 eV belongs to the oxidation state of V^5+^. According to the XRD and Raman signals of Figure 1 and Figure 2, V_2_O_5_ was not observed. Generally, the surface free energy of the film enhances the activity of surface atoms, and the oxidation state of V^5+^ is more stable than V^4+^. In combination with previous reports [26,27], we believed that V^5+^ formed by the spontaneous surface oxidation of VO_2_ stored in the air.

### 3.2. Variation of VO_2_/6H-SiC Thin Films Resistance in S_1_ and S_2_ Groups During MIT

Figure 5 shows resistance-temperature curves for all VO_2_/6H-SiC thin films. Figure 5a–d corresponds to S_1-1_, S_1-2_, S_1-3_ and S_1-4_, respectively. The resistance variations in the samples of S_1-1_, S_1-2_ and S_1-3_ were 3–4 orders of magnitude, and the phase transition temperatures were at about 69 °C. However, it can be seen from Figure 5d that the S_1-4_ sample does not show resistance-temperature curves of a typical VO_2_ films with temperature change. The appropriate substrate temperature can promote the epitaxy growth of thin films on the substrate. During the preparation of the S_1-4_ sample, the high substrate temperature causes a large amount of reflection during the deposition of VO_2_ into the substrate, thus forming discontinuous islands, as shown by the FE-SEM image of the sample.

In order to more clearly reflect the phase transition temperature of the sample, d(R)/d*T* curves for the samples of S_2-1_, S_2-2_, S_2-3_ and S_2-4_ were presented, respectively. According to the resistance-temperature curves of the samples in the S_2_ group, the phase transition temperatures of all VO_2_/6H-SiC films were in the range of 69–70 °C, which is consistent with the previous report by Zhou, et al. [28]. The reason for the higher phase transition temperature of the sample is the slight lattice mismatch between the VO_2_ films and 6H-SiC substrates. We calculated the resistance change and the hysteresis width (defined as Tc(heat)–Tc(cool)) of the derivative curve in the MIT process for the samples in the S_2_ group, and listed the data in the following Table 1:

Obviously, it can be seen from Figure 6a–d that the resistance corresponding to the VO_2_ (R) film in the S_2_ group decreases gradually with the increase of deposition time. Figure 3e–h shows the change trend of surface continuity of the VO_2_ films with the increase of deposition time. In fact, integrated surface is facilitated to the conduction of free electrons on the VO_2_ (R) films. As a result, the variation multiples of the phase change resistance of the sample in the S_2_ group increased gradually. It is worth noting that the amplitude of the resistivity ratio during the MIT of VO_2_ film with optimized condition is up to 5.66 × 10^4^, and the corresponding hysteresis width was only 3.3 °C. The excellent MIT characteristics of VO_2_/6H-SiC films are better than the growth of VO_2_ film on other substrates, such as p-GaN/sapphire [29], Si [30], ZnO [31] and sapphire substrates [32].

Based on the above discussion, we believed that the appropriate substrate temperature and deposition time were critical for growing high quality VO_2_ films. On the one hand, the lower substrate temperature was not conducive to the growth of VO_2_ films on the substrate surface; on the other hand, the high substrate temperature increases the reflection of VO_2_ molecules deposited onto the substrate surface, resulting in the inability to the formation of highly continuous films. With the increase of deposition time, meanwhile, the VO_2_ grain size on the surface of the substrate becomes larger, and the grain boundary decreases, so that the compactness of the film continuously increased. VO_2_ films with high continuity were more effective to the conduction of electrons, which shrinking the hysteresis width in the process of MIT and reducing the resistance of VO_2_ (R). The interface between VO_2_ (020) and 6H-SiC (0006) caused the stress of VO_2_ film. However, according to the variation trend of VO_2_ (020) diffraction peak in Figure 1b, it could be clearly concluded that with the prolongation of VO_2_ deposition time, the stress of the film was relaxed, which can improve the phase transition characteristics of VO_2_ films.

## 4. Conclusions

VO_2_ thin films with excellent phase transition properties were successfully prepared on 6H-SiC (0001) substrate. Optimized substrate temperature and extended deposition time of VO_2_ particles can effectively improve the quality of the samples and help to reduce the resistance of VO_2_ in the R phase, which was as low as 13.6 Ω. During the MIT of VO_2_/6H-SiC film, the maximum amplitude of resistivity change was 5.66 × 10^4^, and the width of the resulting hysteresis loop was only 3.3 °C. This ultra-high quality VO_2_/6H-SiC (0001) thin film preparation was attributed to the good lattice matching as well as the high thermal conductivity properties between the substrate and the epitaxial film. The wide bandgap and great compatibility with metal oxide semiconductors make the combination of 6H-SiC as well as VO_2_ films helpful in exploring new MIT-related finite components.

## Figures and Tables

**Figure 1 nanomaterials-09-01061-f001:**
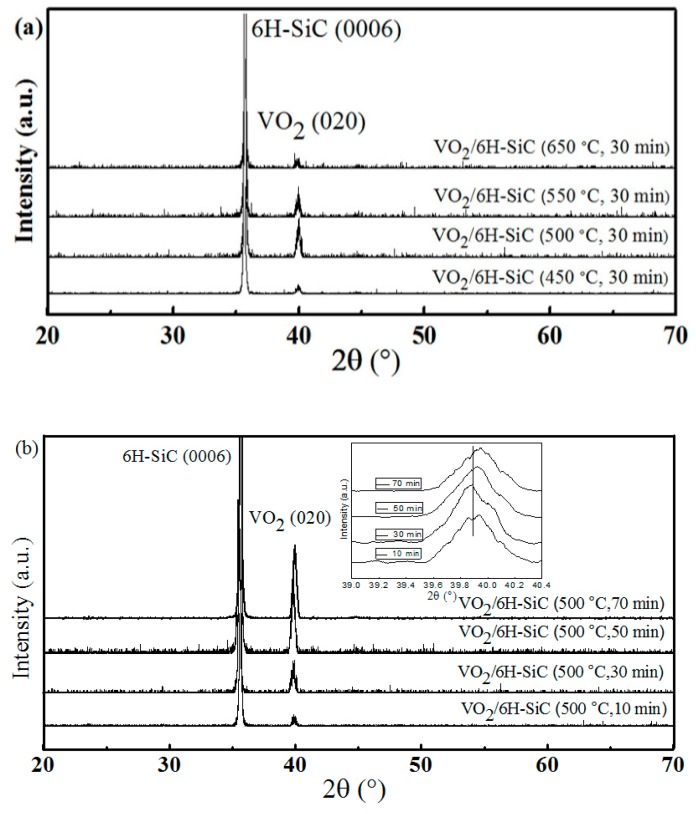
(**a**) and (**b**) were X-ray diffractometer (XRD) patterns of the samples of S_1_ and S_2_ group, respectively. Inset: enlarged view of the corresponding (020) diffraction peak of the VO_2_/6H-SiC films at different deposition times.

**Figure 2 nanomaterials-09-01061-f002:**
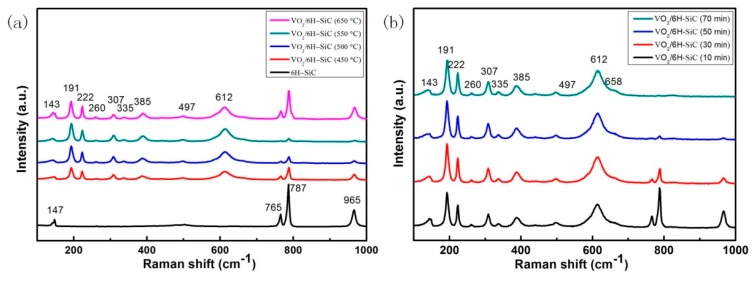
(**a**) Raman spectra of the VO_2_/6H-SiC thin films prepared at different substrate temperatures (450 °C, 500 °C, 550 °C, 650 °C); (**b**) Raman spectra of VO_2_/6H-SiC thin films prepared at different deposition time (10 min, 30 min, 50 min, 70 min) while the substrate temperature remains unchanged at 500 °C.

**Figure 3 nanomaterials-09-01061-f003:**
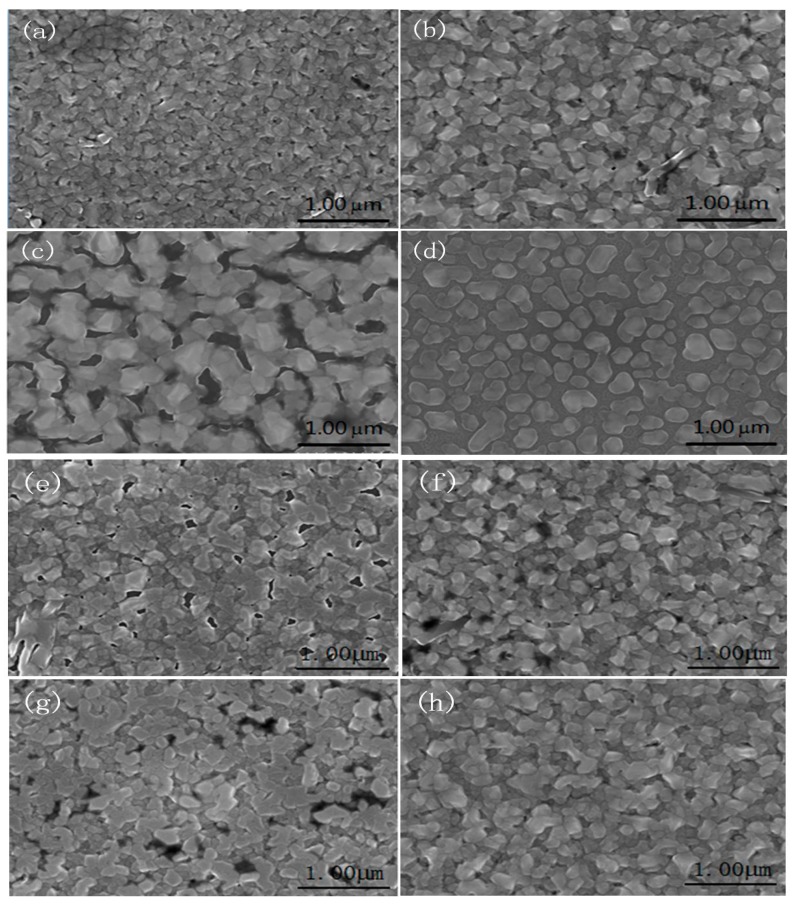
Representative Field Emission Scanning Electron Microscopy (FE-SEM) images of VO_2_/6H-SiC films, (**a**–**d**) corresponds to SEM images from S_1*-*1_ to S_1*-*4_; and SEM images displayed by (**e**–**h**) were sequentially assigned to the samples in the S_2_ group with deposition time of 10 min, 30 min, 50 min, and 70 min.

**Figure 4 nanomaterials-09-01061-f004:**
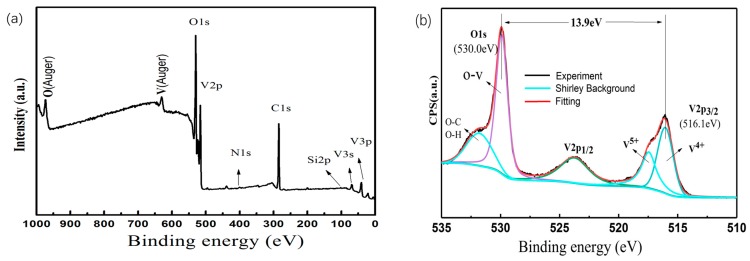
(**a**) XPS measure spectra with binding energy in the range of 0–1000 eV for sample S_2-2_. (**b**) The enhanced high-resolution spectra with binding energy in the range of 510–535 eV and the fitting results for sample S_2-2_.

**Figure 5 nanomaterials-09-01061-f005:**
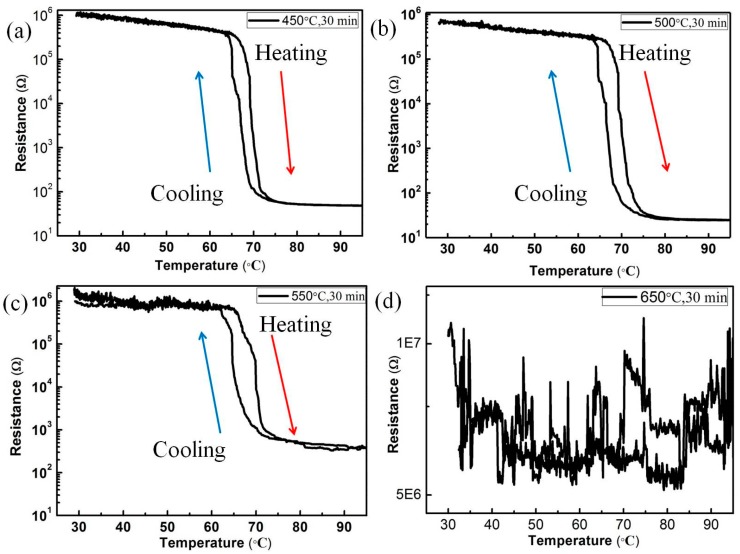
Thermal hysteresis loops of sheet resistance of the samples of S_1_ group, (**a**)–(**d**) corresponding substrate heating temperatures are 450 °C, 500 °C, 550 °C and 650 °C.

**Figure 6 nanomaterials-09-01061-f006:**
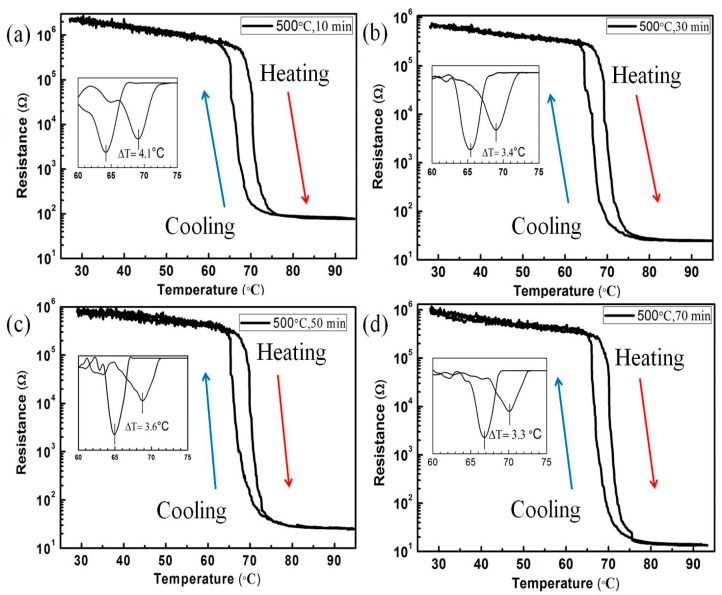
Temperature dependent resistivities of the samples of S_2_ group, (**a**)–(**d**) corresponds to the deposition time of 10 min, 30 min, 50 min and 70 min, respectively. The inset showed the related derivative plot of resistance vs. temperature for VO_2_ film on 6H-SiC.

**Table 1 nanomaterials-09-01061-t001:** The resistance change and the hysteresis width of the samples in the S_2_ group.

S2 Group	Resistance Change	Hysteresis Width (°C)
S_2-1_	2.60 × 10^4^	4.1
S_2-2_	2.71 × 10^4^	3.4
S_2-3_	3.68 × 10^4^	3.6
S_2-4_	5.66 × 10^4^	3.3

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
