# Peer review of "Enhanced Phase Transition Properties of VO2 Thin Films on 6H-SiC (0001) Substrate Prepared by Pulsed Laser Deposition"

_nanomaterials, 2019, doi:10.3390/nano9081061_

Round 1

Reviewer 1 Report

I would like to recommend this paper for publication, due to the careful study presented on high quality epitaxial VO2 thin films prepared by pulsed laser deposition. However, I would suggest to better specify the choice of 6H-SiC substrates. This will justify the importance of the work highligthing its novelty.

Reviewer 2 Report

This manuscript contains the interesting matter. I recommend the publication after revision as following:

1.      Title: VO2/6H-SiC (0001) thin films -> VO2 thin films on 6H-SiC (0001) substrate

2.      There is no information about thickness of VO2 thin film.

3.      How is 500 celsius 70 min ?

4.      Mentioning the application of VO2 thin film is better.

5.      Line 50-51: In spite of “adjustable phase transition, I think that it can be controlled. In Fig. 5,6, transition temperature is fixed around at 70 celsius.

Reviewer 3 Report

The paper presents new results for VO2 films. Results are well presented.

Minor language and other corrections required, e.g.,

Line 35: delete „While“

Line 44: replace „growth“ with „grown“.

Line 55: use “5 mm x 5 mm x 0.5 mm”

Line 102: …. results “are” shown.

Line 142: Figures 3 “are” FE_SEM images …

Line 154: The XPS spectrum of …. “is” shown in Figure 4a.
